

# The 4.2 ka BP event in the Central Mediterranean: New data from Corchia speleothems (Apuan Alps, central Italy)

Ilaria Isola[1], Giovanni Zanchetta[1,2], Russel N. Drysdale[3], Eleonora Regattieri[2,3], Monica Bini[1,2], Petra Bajo[4], John C. Hellstrom[4], Ilaria Baneschi[5], Piero Lionello[6], Jon Woodhead[4], and Alan Greig[4]

[1]Istituto Nazionale di Geofisica e Vulcanologia, Pisa and Rome, Italy
[2]Dipartimento di Scienze della Terra, University of Pisa
[3]School of Geography, University of Melbourne, Australia
[4]School of Earth Sciences, University of Melbourne, Australia
[5]Istituto di Geoscience e Georisorse-CNR, Pisa, Italy
[6]Dipartimento Scienze e Tecnologie Biologiche e Ambientali, Università del Salento, Lecce Italy

**Correspondence:** Ilaria Isola (ilaria.isola@ingv.it)

**Abstract.** We present new data on the 4.2 ka BP event in the central Mediterranean from Corchia Cave (Tuscany, central Italy) stalagmite CC27. The stalagmite was analysed for stable isotopes ($\delta^{13}$C and $\delta^{18}$O) and trace elements (Mg, U, P, Y), with all proxies showing a coherent phase of reduced cave recharge between ca. 4.5 and 4.1 ka. Based on the current climatological data on cyclogenesis, the reduction in cave recharge is considered associated to weakening of the cyclone centre located in the Gulf
5 of Genoa in response to reduced advection of air masses from Atlantic during winter. These conditions, which closely resemble a positive North Atlantic Oscillation (NAO)-like configuration, are associated with cooler and wetter summers, with reduced sea warming, which reduced the western Mediterranean evaporation during autumn-early winter further reducing precipitation.

# 1 Introduction

10 A major and much-discussed example of a potential global "megadrought" and cooling during the Holocene occurred between ca. 4.2-3.9 ka BP (4.2 ka BP event hereafter) (Weiss, 2015, 2016). One of the best-documented case studies of the occurrence of the 4.2 ka BP event in the Mediterranean basin comes from the RL4 flowstone from the Renella Cave (Drysdale et al., 2006; Zanchetta et al., 2016). The Mg\Ca molar ratio, $\delta^{13}$C and organic matter florescence records obtained from RL4 flowstone calcite indicated a prominent reduction in cave recharge between ca. 4.3 and 3.8 ka, corresponding to a pronounced drier
15 period shown by the $\delta^{18}$O record. This was somewhat surprising considering the cave's geographic position. Renella Cave is located in a narrow valley draining the western side of Apuan Alps (Fig. 1), a mountain belt that receives its precipitation from the air masses of North Atlantic origin (Reale and Lionello, 2013) that interact with the most important cyclogenesis centre of the Mediterranean region, the Gulf of Genoa (Fig. 1). The Gulf of Genoa cyclogenesis is most active between November and February (Trigo et al., 2002), but is a persistent feature over the whole year (Lionello et al., 2006). To a large extent, its

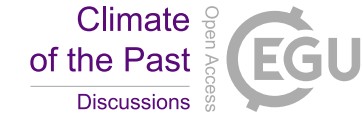

location is determined by local topography, with the Alps (but also the Apuan Alps and Apennines, which bound the eastern side of the Gulf of Genoa) playing a major role by trapping air masses moving eastward and triggering Genoa cyclones (Trigo et al., 2002). The combination of these factors produces high precipitation amounts, locally reaching values up to 3000 mm\yr over the Apuan Alps (Rapetti and Vittorini, 1994). Cyclogenesis in the Gulf of Genova seems to be further maintained by

cold advection of air masses on the western flank of the larger synoptic system towards the relatively warm temperatures over the Mediterranean Sea. Numerical experiments indicate that this is not a triggering factor, since Genoa cyclones are mainly topographically induced. It may, however, be crucial for determining the maximum intensity reached by the local low pressure systems (e.g., McGinley and Zupanski, 1990; Trigo et al., 2002). The finding of a reduction of cave recharge over Renella Cave at the time of the 4.2 ka BP event implies a substantial reduction in the advection of moisture from North Atlantic and

the cyclogenesis over the area, despite the fundamental orographic role exerted by the Apuan Alps. In this paper, we further explore the 4.2 ka BP event in the Apuan Region, presenting new data (stable isotope, and trace elements, Mg, P, Y and U) from a Corchia Cave speleothem. Corchia Cave is located at a higher altitude compared to Renella Cave and it is characterized by a deeper and more complex plumbing system (Drysdale et al., 2004, 2006; Piccini et al., 2008; Zanchetta et al., 2007, 2014).

## 2   Site description

The Apuan Alps comprises intensively karstified marbles and metadolostones bounded by a Paleozoic basement mostly formed by phyllite, the latter considered substantially impermeable (Ciarapica and Passeri, 1994). The cave has been described in detail elsewhere (Drysdale et al., 2004; Piccini et al., 2008; Baneschi et al., 2011) and only general information is reported here. The stalagmite object of this study (stalagmite CC27) was collected from the "Galleria delle Stalattiti" (GdS), which is located ca. 400 m below the surface, at ca. 840 m a.s.l. The chamber has a constant mean annual temperature of 8.4 °C and receives a

recharge of 2500–3000 mm\yr over an elevation range of ca. 1200–1400 m (Drysdale et al., 2004; Piccini et al., 2008). Dripwaters in the chamber have a near-constant oxygen isotope composition ($\delta^{18}$O: ca. -7.4‰, Piccini et al., 2008; Baneschi et al., 2011), which is consistent with predicted averaged $\delta^{18}$O values of rainfall at this estimated recharge elevation range (Mussi et al., 1998; Drysdale et al., 2004). The carbon isotope composition ($\delta^{13}$C) of dissolved inorganic carbon (DIC) is similarly constant (ca. -4‰, Baneschi et al., 2011), and reflects the low contribution of biogenic $CO_2$ due to the thin vegetation cover,

the long interaction with the marble bedrock and, possibly, shifts between closed vs open-system condition and sulphuric-acid dissolution (Bajo et al., 2017). Several drips and pool waters analyzed in the GdS show very constant values for pH, ion concentrations and isotopic composition, suggesting well-mixed waters and a stable, deep plumbing system (Baneschi et al., 2011).

## 3   Material and methods

CC27 is a 20 cm-tall stalagmite collected in situ in the lower part of the chamber. It was sectioned, polished and sampled along the growth axis for stable isotope analysis using a milling machine with a 1 mm-diameter drilling bit, at 0.6 mm increments.





Sampling resolution was increased to 0.2 mm between 58.3 and 64.7 mm from top. Isotope ratios were measured using a GV Instruments GV2003 continuous-flow isotope-ratio mass spectrometer at the University of Newcastle, Australia. All results are reported relative to the Vienna Pee Dee Belemnite (V-PDB) international scale. Sample results were normalized to this scale using an internal standard (NEW1) previously calibrated using the international standards NBS-18 and NBS-19. Analytical

uncertainty for $\delta^{18}O$ and $\delta^{13}C$ was 0.09‰ and 0.05‰ respectively. Trace element analyses were carried out at the School of Earth Sciences, The University of Melbourne, using a 193 nm ArF excimer laser-ablation system coupled to an Agilent 7700xquadrupole ICPMS. The Helex laser-ablation system (Woodhead, 2007) is driven by the GeoStar software (Resonetics). Prior to trace element determination, the sample was twice pre-ablated to clean the surface using a circular spot of 260 $\mu$m diameter at a scan speed of 500 $\mu$m\min and a laser pulse rate of 15 Hz. Element concentrations were measured from the

pre-ablated surface using line scans parallel to the stalagmite growth axis. A main scan was obtained with a spot size of 55 $\mu$m, a scan speed of 50 $\mu$m\s and a laser pulse rate of 10 Hz. To check for lateral data consistency two additional lower-resolution scans were performed 500 $\mu$m apart and parallel to the main scan line. Quantification was carried out using the NIST SRM612 glass reference as external standard. The Standard was analysed three times with the same spot size and a scan speed of 15 $\mu$m\s. Raw mass spectrometry data were reduced using Iolite software (Hellstrom, 2008) and data were internally normalised to

$^{43}$Ca. Eleven powder samples were taken for U-Th dating, which were performed at the University of Melbourne (Australia) by a Nu Instruments MultiCollector Inductively Coupled Plasma–Mass Spectrometer (MC-ICP-MS), according to the analytical method described in detail by Hellstrom (2003). The age model was obtained with the Stal-Age software (Scholz and Hoffman, 2011). Here we present the data concerning the top part (125 mm) of the CC27 stalagmite.

## 4 Results

The corrected U-Th ages range from 2.67 ± 0.70 ka and 7.199 ± 0.11 ka (Table 1). Almost all ages are in stratigraphic order within the associated uncertainties, only one age was rejected as outlier. The calculated age-depth model ranges from 7.368 to 2.437 ka (Fig. 2). Stable isotope values (average -1.2 ± 0.30‰ and -5.0 ± 0.19‰ for carbon and oxygen respectively) are plotted vs age in Figure 3. $\delta^{13}C$ and $\delta^{18}O$ show a general covariant patterns, with higher values centered at ca. 5.9-5.7 ka, 5.2-5.3 ka, 4.5-4.1 ka, and between ca. 2.8-2.7 ka. In the rest of the paper, we will focus on the 4.5-4.1 ka interval, which

appear the most prominent and chronologically equivalent to 4.2 ka BP event (Fig. 3). Figure 3 shows also the Mg, U, P and Y elemental concentrations plotted versus age. In Table 2, the Pearson r correlations coefficients between different trace element pairs are reported. Isotope ratios and Mg are well-correlated (Fig. 3), with most of the lower isotope values corresponding to lower Mg values and vice versa. U displays an opposite behavior, with a significant negative correlation with stable isotope values and Mg content (r=-0.85; Table 2). The P and Y concentrations show a general positive correlation (r=0.60; Table 2)

and are weakly, though significantly, positively correlated with U (r=0.32 and r=0.29 for P and Y respectively, Table 2) and negatively correlated with Mg (r=-0.39 and -0.33 for P and Y respectively, Table 2). Both P and Y show the lowest values between 4.5-4.1 ka as well (Fig. 3). All values are significant at P <0.05.





## 5 Discussion

### 5.1 Speleothem stable isotopes composition

$\delta^{18}$O values of calcite ($\delta^{18}$O$_c$) in the Apuan speleothems have been mostly interpreted on the premise that the $\delta^{18}$O in precipitation ($\delta^{18}$O$_p$) in the western Mediterranean is dominated by the "amount effect", i.e. a positive shift in $\delta^{18}$O$_p$ values corresponds to a lower precipitation amount and vice versa, with a gradient of ca. -2‰\100 mm\month precipitation (Bard et al., 2002). On the other hand, the temperature effect on precipitation (ca. +0.3 ‰; Bard et al., 2002) is almost equal but opposite in sign to the effect of a change in temperature on water-calcite isotopic fractionation (-0.24‰; Kim and O'Neil, 1997). Thus, as a first order interpretation, higher $\delta^{18}$O$_c$ values are considered to indicate a shift towards drier conditions while lower values are associated with more humid periods (Drysdale et al., 2004; Zanchetta et al., 2007; Regattieri et al., 2014a). This "paradigm" holds for central and eastern Mediterranean speleothems (e.g., Bar-Matthews et al., 2003; Regattieri et al., 2018; Finné et al., 2017) and is utilized also for interpreting lacustrine marls, land snail shells and pedogenic carbonates in the region (e.g., Colonese et al., 2007, 2014; Regattieri et al., 2015, 2017; Borretto et al., 2017; Zanchetta et al., 2007, 2017). However, this picture can be complicated by changes in the seasonality of rainfall distribution. For instance, Longinelli (2006) observed that years with an anomalous reduction of precipitation from spring to late autumn may have annual average isotopic values lower than expected, adding weight to the prevailing winter signal ( i.e. more negative $\delta^{18}$O of precipitation, Dansgaard, 1964). This anomaly can be transferred to continental carbonates (Regattieri et al., 2015), and especially to Corchia speleothems, because cave recharge is mostly related to autumn-winter precipitation (Piccini et al., 2008). It has also been noted that changes in the isotopic composition of the source of vapor (Atlantic Ocean vs Mediterranean Sea and changes in isotopic composition of the source) may have an additional effect (Bar-Matthews et al., 2000; Marino et al., 2015; Rohling et al., 2015). Despite these potential uncertainties, the interval with higher $\delta^{18}$O values at 4.5 - 4.1 ka, should primarily indicate a reduction in cave recharge induced by a decrease in local precipitation, most probably related to winter, i.e. the season of maximum cave recharge (Piccini et al., 2008). This may also indicate a change in the proportion of cave recharge between the cold (autumn-winter) and the warm (spring-summer) half of the year. In the western Mediterranean, and in temperate settings more generally, speleothem $\delta^{13}$C composition arises from the relative contribution of $^{13}$C-enriched $CO_2$ derived from bedrock dissolution and of $^{13}$C-depleted $CO_2$ deriving from biological activity in the soil (e.g., Genty et al., 2001; Fairchild and Baker, 2012). Higher soil $CO_2$ production occurs under wetter and warmer conditions, thus $\delta^{13}$C values usually show a general negative correlation with the amount of rainfall, similar to the relationship already discussed for the oxygen. Therefore, the concomitant increase in $\delta^{13}$C between 4.5 and 4.1 ka corroborates the reduction in cave recharge associated with lower biological activity in the soil in the catchment, increasing residence time in the aquifer and increasing the dissolution of bedrock, including by sulfide dissolution (Bajo et al., 2016).

### 5.1.1 Trace elements

Trace element data supplies a fundamental additional control in the interpretation of secular variations in speleothem $\delta^{18}$O composition (e.g., Fairchild and Treble, 2009; Griffiths, 2010), which can complement and improve our partial knowledge on the long-term processes that drive the evolution of oxygen isotopic composition of meteoric precipitation. Mg is generally



transported to the speleothem in solution and, according to Fairchild et al. (2000), Mg\Ca changes in drip waters are due to four different factors, which may act concomitantly in Mg-rich dolomite bedrock (Fairchild and Treble, 2009; Regattieri et al., 2014b), as in the case of Corchia Cave. First, due to the rate of dolomite dissolution being slower than that of calcite (Fairchild and Treble, 2009), a water-rock interaction longer time would cause higher Mg concentrations in the drips. Second, prior
calcite precipitation (PCP) may be important. This involves calcite precipitation in dewatered fractures along the flow path, upstream of the speleothem. Air present in fractures encourage $CO_2$ degassing, calcite deposition and, indirectly, enrichment of Mg\Ca in the solution. This is because the partition coefficient of Mg <1 (Morse and Bender, 1990). The occurrence of PCP is further highlighted by a positive covariance between Mg and $\delta^{13}C$, because $CO_2$ degassing causes $^{13}C$ enrichment in the solution and then in the speleothem calcite (e.g., Baker et al., 1997). Third, incongruent dolomite dissolution is plausible
given the extensive presence of dolomite beds. Finally, there is the possibility of the selective leaching of Mg (Sinclair, 2011). The cumulative effect of these processes is an overall increase Mg content in speleothem calcite during periods of lower cave recharge (i.e. under drier conditions, Hellstrom and McCulloch, 2000; Fairchild et al., 2006; McDonald et al., 2007; Regattieri et al., 2014b). P and Y are mostly transported as detrital phases (Borsato et al., 2007; Fairchild and Treble, 2009), which are a mixture of mineral particulates and organic colloids produced by the weathering of bedrock and leaching of soil (respectively),
and deposited as microscopic particles concentrated in individual layers of calcite (Frisia et al., 2000) or as macroscopically visible clastic-rich layers, transported especially during flood events (Zhornyak et al., 2011). Entrainment of detrital phases in very deep environment like GdS in Corchia is unlikely, as supported by the very pure calcite of the GdS speleothems (see low values in ratio $^{232}Th/^{230}Th$, table 1). For this reason, in our case the transportation of organic colloids is more likely. Studies of temperate ecosystems indicate that both P and Y in cave drip waters are principally bound to soil organic colloids,
and are indicative of infiltration rates, vegetation decomposition rates and soil development (Borsato et al., 2007; Treble et al., 2003). Thus, higher P and Y concentrations can be related to higher water infiltration rates and more developed soil and vegetation (i.e. warmer and wetter conditions), whereas lower values are indicative of lower infiltration and less vegetation and soil productivity (i.e. drier and colder conditions, Borsato et al., 2007; Fairchild and Treble, 2009; Regattieri et al., 2016).

U concentrations in speleothems can be challenging to interpret. In near-neutral pH cave drip-waters, stable uranyl $UO_2^{2+}$
complexes form with carbonate, phosphate and organics (Osmond and Ivanovich, 1992; Treble et al., 2003). During rock alteration and pedogenesis in oxidising conditions, tetravalent U(IV) changes to hexavalent U(VI), which is soluble in water (Ortega et al., 2005). U concentration in water may be influenced by changes in soil redox conditions. An increase in microbial respiration would lead to reduced oxidation within the soil, and this may cause less uranium to be oxidized to the hexavalent state, and thus to lower uranium concentrations in the seepage water (Osmond, 1980; Hellstrom and McCulloch, 2000). Soil
and groundwater P concentrations may also influence the transport of U through the strong affinity between phosphate and uranyl ions (Treble et al., 2003). After percolation through the karst network, U(VI) precipitates in calcite as $UO_2^{2+}$, probably as a replacement of $Ca^{2+}$. Experimental studies on synthetic calcite show that the partition coefficient of U between solution and solid varies between 0.06 and 1.43 (Day and Henderson, 2013), with no significant relationship with growth and drip rates or temperature. To disentangle the predominant environmental driver of U concentration in CC27, a comparison with the other
proxies is useful. Our time series shows only a weak positive correlation between U and P (Table 2), suggesting that some of the



U may be bound to phosphate and to organic colloids. This is supported by Riotte et al. (2003), whose observations confirmed the possible control of colloids on the transport of U isotopes in freshwaters. A strong negative correlation between U and Mg is also observed (Table 2), whilst the general pattern of U changes appears negative correlated with both $\delta^{13}$C and $\delta^{18}$O (Fig. 3), suggesting that higher\lower U concentration is related to wetter\drier conditions. This can suggest both, a stronger leaching

from rocks overlying GdS and hosting high U contents (e.g. Brecce di Seravezza) or a partition coefficient for U in GdS water >1, in this case changes in U concentration of CC27 could be related to the degree of PCP, as for Mg but in an opposite fashion. Overall the most simple mechanism to control U\Ca ratio would be the cave recharge and consequently leaching mechanism. Variations in soil redox conditions are likely to be negligible due to the thick bedrock cover over GdS. Indeed, the increase in soil biological activity, responsible for less oxidizing conditions and greater colloidal transport, would have been favored by

wetter and warmer conditions, thus affecting U concentrations.

During the 4.2 ka BP period (Fig. 3), lower P and Y likely indicate reduced colloidal production in soil and\or lower infiltration and transport into the cave. This is indicative of reduced vegetation development and water infiltration due to a reduction in precipitation. The latter causing the reduction in U content too. These patterns are in agreement with the concomitant increase of Mg related to the occurrence of PCP, and thus to drier conditions, as testified also by the increase in $\delta^{13}$C, related both to

soil $CO_2$ supply but also to $CO_2$ degassing upstream of feeding drip.

Therefore, the shifts observed in all trace elements (higher Mg, lower U, P and Y) and in both stable isotope composition (higher $\delta^{13}$C, and $\delta^{18}$O), together define a period of reduced cave recharge between 4.5 ka and 4.1 ka (Fig. 3). Assuming that the trace-element behavior is related to hydrological variations as discussed before, we can combine the different individual trace elements records to produce a composite mean anomalies time series (Regattieri et al., 2014b). This approach detects coherent

variability across multiple speleothem properties, reducing the noise inherent in each series and highlighting the environmental signal. To produce a mean anomaly series, the individual records were smoothed using a ten-point moving average, then normalized to produce correspondent time series of anomalies (i.e. deviations from a zero mean expressed in standard deviation units). Because the pattern of Mg is reversed with respect to the other trace elements, the latter were multiplied by -1. Standard scores of the individual series were then averaged for each time increment to produce a composite time series (z score) where

low (high) values correspond to relatively wetter (drier) conditions (Fig. 3). The result reveals a prominent drier interval in the z-score series between ca. 4.5 and 4.1 ka (Fig. 3). This period of drier conditions overlaps, within uncertainties, the drier phases identified in the RL4 flowstone from Renella Cave (Fig. 4, Drysdale et al., 2006; Zanchetta et al., 2016). Figure 4 also shows the comparison between the CC27 multi-proxy record and that of CC26, a previously studied speleothem from the same chamber (Zanchetta et al., 2007; Bajo et al., 2016; Regattieri et al., 2014b, 2017; Tognarelli et al., 2018) Interestingly, the prominent

shift in the $\delta^{18}O_c$ of CC27 is not apparent in the CC26 $\delta^{18}O_c$ record. However, during the 4.2 period, CC26 shows a pronounced increase in the Mg\Ca ratio (Regattieri et al., 2014), consistent with Mg (and other trace elements) variation measured in CC27 record (Fig. 4). The difference between CC26 and CC27 $\delta^{18}O_c$ records deserves further comment. For deep, large caves like Corchia, the complexity of the recharging system leads to the presence of different reservoir compartments, characterized by different infiltration points and by different hydrological routings. In addition, GdS has the peculiarity to receive drip recharge

from lateral paths because the chamber is overlain by the impermeable phyllitic basement (Piccini et al., 2008). This increases



the drip-path complexity. Different paths and mixing of water may create small differences in the geochemistry of different drip points (e.g., Fairchild and Treble, 2009), though located in close proximity (i.e. in the same cave chamber). In addition, the steep slopes of Mt. Corchia can produce important changes in the mean altitudes of infiltration for different drips (up to several hundred meters over a relatively short distance) inducing the "altitude isotope effect" (Dansgaard, 1964). Mussi et al.

(1998) derived an altitudinal effect of ca. -0.15‰\100 m for the western Apuan slopes, which is significant considering the minor isotopic changes recorded in this cave during the Holocene (Zanchetta et al., 2007). Differences in the long-term isotopic records of different speleothems from the same chambers may be related to long-term drip-path evolution and, particularly, to changes in the mean altitude of recharge and\or to mixing of different reservoir compartments having different recharge areas. Another mechanism that can explain the different $\delta^{18}$O trends is precipitation of calcite out of isotopic equilibrium conditions.

Indeed, kinetic fractionation causes unpredictable isotopic enrichment (Mickler et al., 2006). However, this is unlikely for both CC26 and CC27, as they are both composed of monotonous columnar calcite (Bajo et al., 2016), the fabric thought to occur when speleothems are continuously wet, under relatively constant flow and calcite precipitation from fluids at or near isotopic equilibrium (e.g., Frisia et al., 2002; Frisia and Borsato, 2010).

### 5.1.2   Other evidences of environmental changes over the Apennine and central Italy

Environmental and climate change during the drier period identified over the Apuan Alps between ca. 4.5 and ca. 3.8 ka (considering the whole range of ages present at Corchia and Renella caves), was previously reported for several sites on the Apennines and the surrounding coastal plains. In the Central Apennines, geomorphological evidence of cooling and for the re-formation of Calderone Glacier were reported (Giraudi, 2004, 2005), and this event is stratigraphically constrained by the presence of Agnano Mt. Spina (ca. 4.4 ka cal BP) and Avellino (ca. 3.8 ka cal BP) tephra layers (Zanchetta et al., 2012,

2018). This suggests a reduction of either summer and\or winter temperature, that may have caused a prevalence of snow precipitation during winter, higher winter accumulation and\or a reduced summer ablation. Cooler conditions during summer are also supported by fossil chironomid assemblages in the nearby Northern-Apennine Verdarolo and Gemini lakes (Fig. 5, Samartin et al., 2017). In further geomorphological evidence, a period of reduced alluvial activity is recognizable over the Northern Apennines between ca. 4.4 to 3.6 cal ka BP (Giraudi, 2014), consistent with drier conditions or rainfall patterns

less prone to produce alluvial layers. Further, a general phase of reduced valley sedimentary infilling occurs between ca. 4.2 and 3.6 cal ka BP in the upper Turano River drainage basin, 60 km northeast of Rome, although the origin of this phase is interpreted to be related to a general phase of biostasy related to a well-developed vegetation cover preventing runoff and sediment transport (Borrelli et al., 2014). At lower altitudes, stratigraphic data from the Pisa alluvial plain (Fig. 1) indicate river channel avulsion (Sarti et al., 2015) during this period, suggesting that occasionally some large floods may have occurred.

On the contrary, Piccarreta et al. (2011) show an increase of alluvial events between ca. 4.3 and 4.1 cal ka BP in some valleys of Basilicata (Southern Italy). Quantitative reconstruction of climate parameters using pollen concentrations from Accesa Lake (southern Tuscany, Fig. 1 and Fig. 5, Peyron et al., 2013) shows reduced mean annual temperature over the considered period. Precipitation, reconstructed from the same record, shows high variability but no significant variations in mean annual values (Peyron et al., 2013). In terms of seasonality, the Accesa Lake records show a winter precipitation increase between



4.4 to 4.1 cal BP and a decrease between 4.1 to 3.7 ca BP, and as well as opposing summer precipitation patterns over both intervals (Fig. 5). The winter precipitation reconstruction from Accesa may thus not be in agreement with proxy data from Corchia speleothems, which indicate a reduction in cave recharge. Overall, Apennines and central Italy records show some contrasting patterns. However, the interpretation of some proxies of environmental change for this region during the Mid-

to Late Holocene needs care for the potential human impact (Zanchetta et al., 2013). Indeed, this period corresponds to the beginning and definitive affirmation of the Bronze Age in Italy (Bietti Sestieri, 2010). The strong impact on the environment of the Bronze Age population makes it mandatory to consider each recorded environmental change as the complex result of both human activity and climate variability. This is particularly true for pollen records (Roberts, 2004; Fyfe et al., 2015), as Bronze Age communities strongly modified the composition of the vegetation. According to Borrelli et al. (2014), many Apennine

catchments underwent a shift from generally prevailing biostasy to conditions of anthropic rhexistasy starting at around 4.2 ka BP.

### 5.1.3 Regional comparison

Looking at the wider regional palaeoclimate, both Mg\Ca and alkenone sea-surface temperature (SST) data from the Alboran Sea, (core ODP-976, SW Mediterranean, Fig. 5, Jiménez-Amat and Zahn, 2015; Martrat et al., 2014), show a temperature

decrease around 4 ka cal BP. In the Gulf of Lyon (NW Mediterranean, Fig 1), the alkenone SST record from KSGC31 core shows marked oscillations with minimum temperature values around 3.9 ka cal BP (Jalali et al., 2016). Interestingly, and also from the Gulf of Lyon, a record of storm activity shows a period of increased storminess between 4.4 and 4.05 ka cal BP (Sabatier et al., 2012). High storm activity in the Gulf of Lyon has been proposed to be related to cooling over the North Atlantic and the western Mediterranean (Sabatier et al., 2012).

### 5.1.4 Synoptic atmospheric conditions

We have already discussed that speleothems over the Apuan Alps show consistent evidence of reduced cave recharge in a period ranging from ca. 4.5 to 3.8 ka. This implies a reduction of cyclones of Atlantic origin (Reale and Lionello, 2013) and of secondary cyclogenesis over the Gulf of Genoa, considering the relation between cyclogenesis and rainfall in the area (Lionello et al., 2006; Reale and Lionello, 2013). A reduction in local cyclogenesis can be associated with a decrease in the arrival of

North Atlantic air masses to Gulf of Genoa and also by reduced transit of North Atlantic cyclones (Reale and Lionello, 2013) and reduction of vapour advection from western Mediterranean, which acts as a moisture source for the central Mediterranean (Nieto et al., 2010). An important point to consider, however, it is not just a reduction in the cyclones reaching the Apuan Alps but also a reduction of precipitation associated with each single cyclone. Model simulations show that projected Mediterranean precipitation reduction in winter is strongly related to a decrease in the number of Mediterranean cyclones, but local changes in

precipitation generated by each cyclone is also important (Zappa et al., 2015). For the central Apennines, winter precipitation is negatively correlated with the North Atlantic Oscillation (NAO) index (López-Moreno et al., 2011), with a negative NAO index associated with elevated precipitation during winter months. This is due to the lower pressure gradient between the Azores High and the Icelandic Low, which causes a southward shift in westerly trajectories, leading to increased penetration of moist air



masses from the Atlantic to the Mediterranean, and stimulating local cyclogenesis. The Genoa GNIP station shows a significant negative correlation between the amount of winter precipitation and the NAO index (r = -0.74). At the same station, a significant positive correlation is found between $\delta^{18}O_p$ and NAO index (r=0.41) (Baldini et al., 2008). This is in agreement with rainfall amount and seasonality effects, as enhanced winter precipitation during NAO phases should be characterized by lower $\delta^{18}O$

values. In central Europe, where the amount effect is negligible, a relatively strong positive impact of the winter NAO index on precipitation $\delta^{18}O$ has been attributed to the higher frequency of cold easterly winds carrying $^{18}O$-depleted moisture during NAO-negative phases compared to warmer westerly winds, which carry $^{18}O$-enriched moisture from the North Atlantic Ocean and Mediterranean Sea into central Europe during positive NAO winters (Baldini et al., 2008). However, persistent penetration of easterly moisture sources in the Gulf of Genoa is prevented by the Alpine ridge and there is a more important contribution

of local recycling of vapour from the Mediterranean Sea (Baldini et al., 2008). Thus, the correlation observed at the Genoa GNIP station suggests that during periods of reduced North Atlantic moisture (i.e. NAO-positive phase causing northward shift of storm-track trajectories), precipitation from vapour of Mediterranean origin may dominate locally. The higher $\delta^{18}O$ values should be in response to the effect of lower rainout of vapor masses (Celle-Jeanton et al., 2004) and the higher seawater isotopic composition of the Mediterranean compared to the North Atlantic (Pierre, 1999). From this point of view, the drier

period recorded in the Apuan speleothems can be related to persistent positive NAO-like conditions, as invoked for some centennial to multidecadal scale climatic changes over the western Mediterranean (e.g., Fletcher et al., 2013; Zanchetta et al., 2014), including during the Medieval Climate Anomaly (Trouet et al., 2009). However, the reconstructed NAO index for the studied interval does not show a pronounced "positive-mode" (e.g., Jackson et al., 2005; Olsen et al., 2012) and thus this mechanism is not an entirely satisfactory explanation (Fig. 5). Evidence from the tropics suggests that, at the time of 4.2 ka BP

event, there was a near-synchronous disruption of the Indian and African Summer Moonson systems, (Fig. 5, Dixit et al., 2014; Welc and Marks, 2014) resulting from a southward shift of the Intertropical Convergence Zone (ITCZ) (Welc and Marks, 2014; Railsback, 2018). The position of the ITCZ and the resulting strength of the boreal summer monsoon both influence summer aridity and temperature in the Mediterranean region, and are driven by atmospheric subsidence and strengthening of the summer high-pressure systems over the basin (Alpert et al., 2006; Eshel, 2002; Gaetani et al., 2011). A southward shift in

the ITCZ can weaken summer aridity, producing milder (cooler) summer conditions over the Mediterranean. This is consistent with the precipitation reconstruction from Accesa Lake (Peyron et al., 2013), which suggests increasing rainfall during summer, and with a temperature reconstruction from chironomids indicating a reduction in summer temperature. On the basis of these observations we suggest that a weakening of high-pressure cells over the western Mediterranean during summer as the ITCZ shifts southward may have had a pronounced effect on Mediterranean Sea temperatures. Reduced seawater temperature at

the end of summer can reduce evaporation during autumn-winter in the western\central Mediterranean Sea, causing reduced advection towards the central Mediterranean (Nieto et al., 2010) and decreased cyclogenesis in the Gulf of Genoa. This effect cannot be completely captured from available SST data, which record spring and\or annual average temperature and not specific late summer-autumn temperature (e.g. Fig. 5, Cisneros, 2016; Jalali et al., 2016). From the available data, it seems possible that a general reduction in precipitation, as reflected in the CC27 stable isotope and trace element data for the Apuan Alps,

must have involved reduced advection of vapour masses during winter and a reduction in cyclogenesis over the Gulf of Genoa.



However, during the summer, tropical weakening of the monsoon systems may have triggered cooler and wetter condition and this may have been one of the causes for the 4.2 ka BP event.

## 6 Conclusions

The stable isotope and trace element composition of stalagmite CC27 from Corchia Cave records reduced moisture in northern Tuscany between 4.5 and 4.1 ka. During this interval, increased $\delta^{13}$C and $\delta^{18}$O values occur at a time when Mg concentrations increase due to PCP and Y and P concentrations decrease due to reduced infiltration, vegetation activity, and soil development in the cave recharge area. Each of these changes is consistent with a shift to drier conditions. The decrease in U concentration can also be ascribed to reduced moisture. This period is, within age uncertainties, in agreement with trace element data from another speleothem from the same chamber (CC26, Regattieri et al., 2014b) and with the multi-proxy record from nearby Renella Cave (Drysdale et al., 2006; Zanchetta et al., 2016), which together provides robust evidence for a regional reduction in precipitation, in spite of some differences in chronology. These results imply a reduction of cyclones originating in the North Atlantic and of cyclogenetic activity in the Gulf of Genoa, probably due to a northward shift in westerly trajectories that reduced the transport of moist air masses from the North Atlantic and western Mediterranean. This suggests a positive NAO-like mode. Based on the available data in the region close to the Gulf of Genoa, a scenario of lower mean annual temperature, reduced precipitation during winter, and cooler and wetter summer conditions appears plausible. This is consistent with a southward shift of the ITCZ during summer, producing a weaker high pressure over western Mediterranean and reduced ocean surface warming which dampens evaporation during autumn-early winter months. These results indicate that the synoptic processes behind the 4.2 ka BP event involved changes not only in average conditions (as reported by the speleothem) but also significant changes at the seasonal scale, which need to be better investigated in future work.




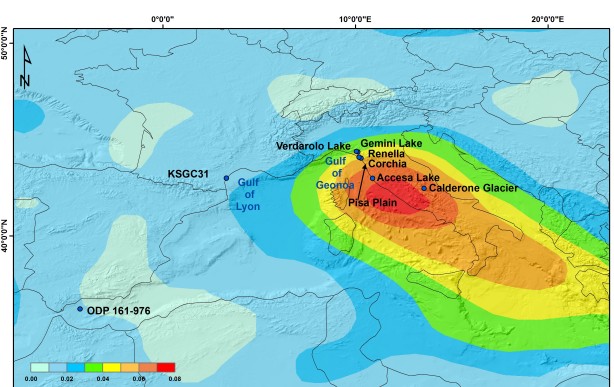

**Figure 1.** Location map of the sites discussed in the text. The figure shows the density track of winter storms according to ERA-interim 1989–2009. Numerical values (number of cyclones\deg2) represent the average spatial density of cyclone centers in the winter season. Only cyclones with a minimum of 1 day duration and 5 hPa depth with respect to the background are included; (modified after Lionello et al., 2012).





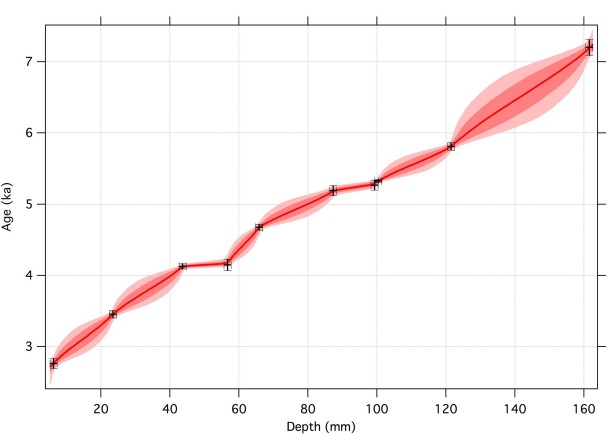

**Figure 2.** CC27 depth–age model. The outer shaded zones define the 95% uncertainties





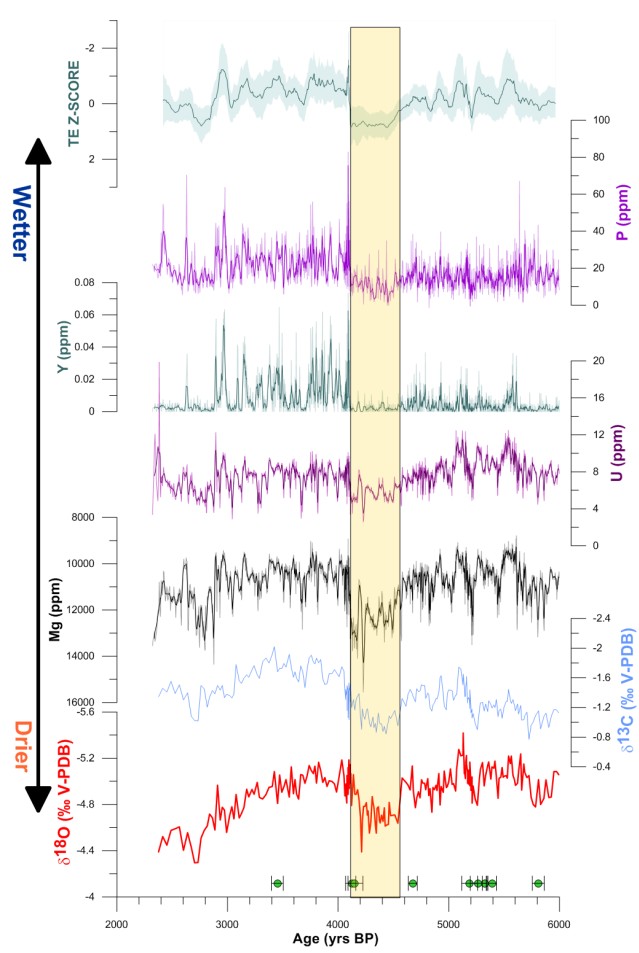

**Figure 3.** Time series of CC27 isotopes and trace elements, from the bottom $\delta^{18}O$, $\delta^{13}C$ , Mg, U, Y, P. Thick lines are 5 pts running average. On the top trace elements Z-score (green line), error is expressed in standard deviation units (light blue strip). Green dots represent U-Th ages, horizontal bars $2\sigma$ errors. Yellow box supposed 4.2 ka cal BP interval as recorded by CC27 proxies.





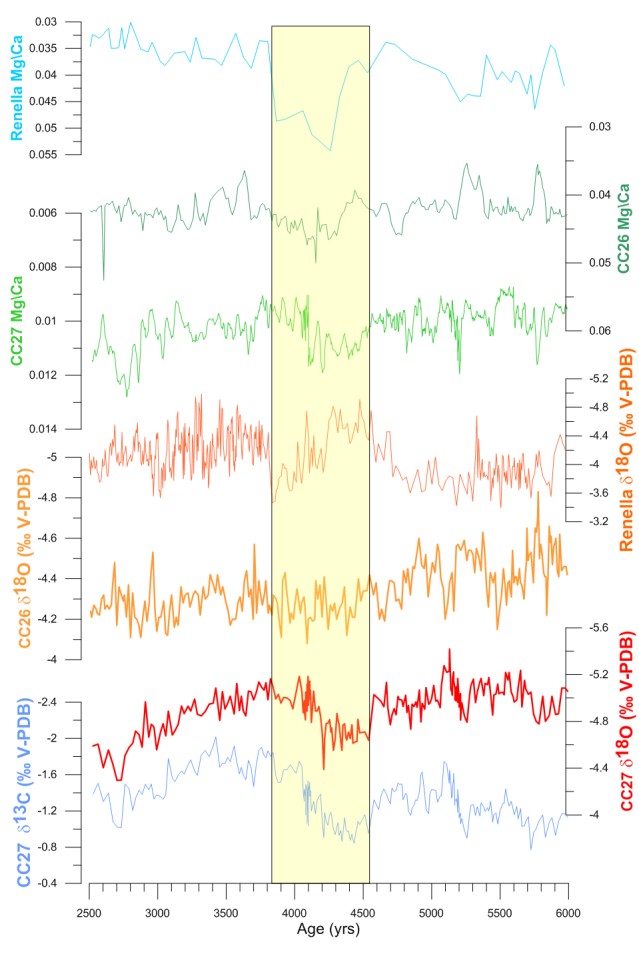

**Figure 4.** Comparison among CC27, CC26 and Renella proxies, form bottom: CC27 $\delta^{18}$O, $\delta^{13}$C, CC26 $\delta^{18}$O, Renella $\delta^{18}$O, CC27 Mg\Ca, CC26 Mg\Ca, Renella Mg\Ca. Yellow box supposed 4.2 ka cal BP interval as recorded by CC27-Renella proxies. Renella $\delta^{18}$O age model is referred to Drysdale et al. in progress, Mg\Ca age model is referred to Drysdale et al. (2006).



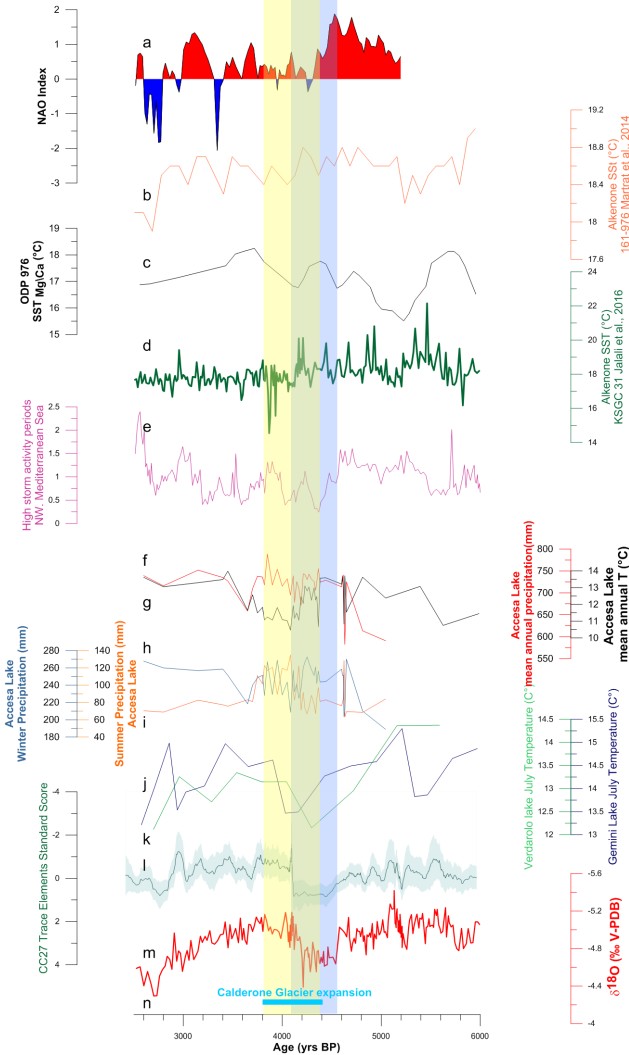

**Figure 5.** CC27 $\delta^{18}$O and trace elements Zscore records compared with paleoclimate regional proxies, from top: a) NAO Index reconstruction (Olsen et al., 2012); b) SST Alkenone temperature record from Alboran Sea (Martrat et al., 2014); c) SST Mg\Ca temperature record from Alboran Sea (Jiménez-Amat and Zahn, 2015); d) Alkenone temperature record from Gulf of Lyon (Jalali et al., 2016); e) Gulf of Lyon Storm activity record (Sabatier et al., 2012); f) Accesa lake mean annual precipitation (Peyron et al., 2013); g) Accesa Lake mean annual temperature (Peyron et al., 2013); h) Accesa lake winter precipitation (Peyron et al., 2013); i) Accesa Lake summer precipitation (Peyron et al., 2013); j) chironomid midges assemblages temperature derived from Gemini Lake fossil (Samartin et al., 2017); k) chironomid midges assemblages temperature derived from Verdarolo Lake fossil (Samartin et al., 2017); l) CC27 trace elements Z-Score; m) CC27 $\delta^{18}$O; n) Calderone Glacier reactivation (Zanchetta et al., 2012).



**Table 1.** U/Th ages for CC27 stalagmite

| Sample ID | Depth (mm) | 230Th/238U | 234U/238U | 230Th/232Th | Uncorrected Age | Corrected Age | 2s |
|---|---|---|---|---|---|---|---|
| CC27 31,32,33 | 6.3 | 0.0162 | 0.6483 | 496.6459 | 2763 | 2761 | 70 |
| CC27-118 | 23.5 | 0.0202 | 0.6507 | 2624.0705 | 3443 | 3450 | 53 |
| CC27-220 | 43.7 | 0.0241 | 0.6517 | 2709.2452 | 4117 | 4124 | 35 |
| CC27-284 | 56.7 | 0.0244 | 0.6538 | 188.9949 | 4178 | 4145 | 77 |
| CC27-330 | 65.9 | 0.0273 | 0.6528 | 398.5427 | 4691 | 4674 | 39 |
| CC27 437 | 87.3 | 0.0302 | 0.653 | 3637.0706 | 5172 | 5188 | 71 |
| CC27-492 | 98.3 | 0.0312 | 0.6518 | 3175.7312 | 5372 | 5392 | 38 |
| CC27 497 | 99.3 | 0.0306 | 0.6522 | 4077.6739 | 5255 | 5266 | 71 |
| CC27-502-503 | 100.4 | 0.0309 | 0.653 | 11788.6375 | 5324 | 5323 | 21 |
| CC27-608 | 121.5 | 0.0337 | 0.6545 | 1613.9824 | 5788 | 5809 | 54 |
| CC27-809 | 161.9 | 0.0415 | 0.6537 | 7342.5966 | 7177 | 7199 | 112 |

Corrected U/Th ages for CC27 stalagmite





**Table 2.** Pearson correlation coefficients r between elements calculated using 10 pt running averages. All values are significant at P <0.05.

| r | P(ppm) | Y(ppm) | U(ppm) |
|---|---|---|---|
| Mg (ppm) | -0.39 | -0.33 | -0.85 |
| P (ppm) | - | 0.60 | 0.32 |
| Y (ppm) | - | - | 0.29 |

20  *Author contributions.*  II, GZ, RD, ER and MB conceived the paper; II and GZ wrote the manuscript, with contribution from RD, ER and MB; II, GZ, RD, PB, JCH, IB, JW and AG contributed for the analytical work; PL contributed for the discussion on climate condition over the area, all the co-authors contributed to the scientific discussion on the conclusion of this paper

*Competing interests.*  No competing interests are present

*Acknowledgements.*  We thank the Federazione Speleologica Toscana and Parco Regionale delle Alpi Apuane for supporting our work on Apuane speleothems. The analytical work presented here has been funded by the Australian Research Council Discovery Project DP160102969 awarded to RD, GZ, ER and JH, and it is part of the ARCA project "Arctic: present climatic change and past extreme events" funded by the Italian Ministry of Education, Universities and Research (MIUR) awarded to II. Part of this study has been developed within the frame of the project, "Climate and alluvial event in Versilia: integration of Geoarcheological, Geomorphological, Geochemical data and numerical simulations" awarded to MB, and funded by the Fondazione Cassa di Risparmio di Lucca. It is also part of "PRA−2018−41 Georisorse e 5  Ambiente" funded by the University of Pisa.





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
