# Peer review of "The 4.2 ka BP event in the Central Mediterranean: New data from Corchia speleothems (Apuan Alps, central Italy)"

_Climate of the Past, 2018_

## Referee Comment (RC1) · Anonymous Referee #1 · 31 Oct 2018

The article by Isola et al. is a nice addition to previous studies that have focused on the 4.2 ka BP event in Central Mediterranean. Past studies on Corchia Cave (mentioned by the authors) have already shown the importance of this location for past environmental reconstructions, justifying the publication of this new and important dataset. The manuscript is well written and well organized. The sections flow logically and the paper captures the reader's attention.

I do have some minor recommendations to improve an already good paper:

1- No picture(s) or clear description of the cave and of the speleothem are given. The authors mention that "The cave has been described in detail elsewhere (Drysdale et

al., 2004; Piccini et al., 2008; Baneschi et al., 2011) and only general information is reported here". That is fine but I would suggest the authors add "informative" pictures of the cave and of the speleothem because, referring to other publications may reduce the impact of this article.

2- A main point: statistical analyses are absent here (except in Table 2) and could have provided a good opportunity to correlate the different signals (especially in case of long-term time-series). I suggest that the authors test their correlations (see Figs 4 and 5) and maybe add information on what they suggested: "robust evidence for a regional reduction in precipitation, in spite of some differences in chronology" (see page 10). I think that the use of statistical analyses would significantly increase the impact of this good article.

3- A last point that needs to be better discussed/tested: the age-model. Clearly, the ages at 98.3 mm (corrected age: 5.392±0.2 ka), 99.3 mm (corrected age: 5266±0.7 ka) and 100.4 mm (corrected age: 5323±0.4 ka) are somewhat ambiguous and need to be discussed in more detail. Even if the "issue" was presented by the authors, I suggest to pay more attention to this because chronology is a key point here and must be unequivocal.

I particularly like two points: 1- "lower mean annual temperature, reduced precipitation during winter, and cooler and wetter summer conditions appears plausible". This is a perfect way to summarize the key findings described in the paper; and 2- "These results indicate that the synoptic processes behind the 4.2 ka BP event involved changes not only in average conditions (as reported by the speleothem) but also significant changes at the seasonal scale". The last sentence summarizes what will probably be a key research avenue for most palaeo-environmentalists during the coming years.

The article could be published as it stands but I strongly encourage its publication after minor revisions.

---

## Referee Comment (RC2) · Anonymous Referee #2 · 8 Nov 2018

The study provides a new data-series from Corchia Cave, focusing on the period of the 4.2 ka BP event. Although Corchia cave, together with Renella Cave, have provided multiple and significant proxies until now, this addition is of great interest since it is appropriate for the description and the exact timing of the aforementioned event, as a matter of resolution and dating. The article is well structured and well written, helping the reader to follow the discussion all along the text. The figures are helpful and well presented. The authoring team has done a good job to provide another useful dataset, this time from the Holocene of the Central Mediterranean. Below my list of recommendations as improvements of the article:

[Figure]

Title: Why 'speleothems'? I was expecting to see more than one after reading the title. Isn't it a bit misleading?

Page 2, line 4 Gulf of Genoa, please follow the same term in whole text

Page 2, Section 2 I find it a bit difficult to follow the site description without being able to visualize the cave setting and the general location in a large scale map. There is no need for a cave description, but I think figure 1 itself cannot help the reader much. I would suggest adding a simple map of the location and the cave setting, incl. the sampling site.

Page 2, Section 3 It is necessary to provide a figure with the speleothem, the subsampling positions for all analyses and the projected growth axis (incl. length or distance from top).

Page 3, Section 2 (line 1-5) Please provide also the preparative method of the stable isotope analysis (eg. acid digestion, including either a citation or a short description with acid type, temperature and duration). Please also provide the carbon and oxygen isotope composition of the internal standard NEW1, including their uncertainties or SD.

Page 5, Section 5.1.1 It is common to use the Mg/Ca ratio in speleothem studies, as the authors did in their previous work as well. Why here only Mg is used, without considering the oscillations of the Ca in these layers? In Fig. 4, we can see Mg/Ca data from CC27 and they look milder as a matter of excursions. To my understanding, Mg/Ca provides a more reliable proxy for interpretation. Nevertheless, Mg/Ca is not discussed, although presented in Fig. 4 (see for instance Page 6, line 31).

Page 6, line 25 '...prominent drier interval'. Here, there is a characterization of a drier event without some supportive remarks. There should be a simple statistical analysis in order to point out the significant events along the CC27 dataset. The statistical analysis can support the interpretation regarding the prominent climatic oscillations/events (for instance, which intervals are prominent indeed and which are not) and help the authors

support their conclusions.

References Please check some citations in the text (for example page 9, line 22 and line 33), where 'et al.' is missing. Also check please the alphabetical order of the reference list (eg. page 21, line 28).

Figure 5. Proxies b, c, j and k datasets are of significantly lower resolution in comparison with the data of CC27 and the purpose of this study (4.2 ka BP event). Do they really help in this figure to make a point out of it?

Based on the recommendations-suggestions above for the improvement of the article, I would suggest the paper to be accepted for publication subject to minor revisions.

———————————————

---

## Author Comment (AC1) · 14 Dec 2018

We thank referee # 1 for the comments and suggestions. Following our replay point by point

"The article by Isola et al. is a nice addition to previous studies that have focused on the 4.2 ka BP event in Central Mediterranean. Past studies on Corchia Cave (mentioned by the authors) have already shown the importance of this location for past environmental reconstructions, justifying the publication of this new and important dataset. The manuscript is well written and well organized. The sections flow logically and the paper captures the reader's attention. I do have some minor recommendations to improve an

already good paper"

We are happy of the positive general comment

"No picture(s) or clear description of the cave and of the speleothem are given. The authors mention that "The cave has been described in detail elsewhere (Drysdale et C1 CPD Interactive comment Printer-friendly version Discussion paper al., 2004; Piccini et al., 2008; Baneschi et al., 2011) and only general information is reported here". That is fine but I would suggest the authors add "informative" pictures of the cave and of the speleothem because, referring to other publications may reduce the impact of this article."

This request is also similar to Referee#2. We have added a new complete figure 2, which shows geology of Mt. Corchia, cave network and a polished section of CC27.

2- A main point: statistical analyses are absent here (except in Table 2) and could have provided a good opportunity to correlate the different signals (especially in case of long-term time-series). I suggest that the authors test their correlations (see Figs 4 and 5) and maybe add information on what they suggested: "robust evidence for a regional reduction in precipitation, in spite of some differences in chronology" (see page 10). I think that the use of statistical analyses would significantly increase the impact of this good article

This is quite similar to one request of Referee#2. We have expanded this point and added a discussion on square-wave plot vs age of index of filtered anomalies of CC27 trace elements, which shows that between ca. 4.5 and 4.1 there is a statistically significant interval interpreted as drier. This approach has been done for CC27 proxies, but also for time-series discussed in the new figure 6. As now explained along the text, for these records we are intended to produce "climatic anomaly" time series irrespective of the specific meaning. From the new figure 6 is possible to observe that between ca. 4.5 and 4.2 ka there is a well expressed "negative" index of filtered anomalies, which roughly implies that this is a tendentially cold, dry and stormy period

3- A last point that needs to be better discussed/tested: the age-model. Clearly, the ages at 98.3 mm (corrected age: 5.392±0.2 ka), 99.3 mm (corrected age: 5266±0.7 ka) and 100.4 mm (corrected age: 5323±0.4 ka) are somewhat ambiguous and need to be discussed in more detail. Even if the "issue" was presented by the authors, I suggest to pay more attention to this because chronology is a key point here and must be unequivocal.

Yes, these ages comprised in 3 mm are indistinguishable within the age error. We have inserted a more sound discussion on the age model and as it has been statistically obtained.

I particularly like two points: 1- "lower mean annual temperature, reduced precipitation during winter, and cooler and wetter summer conditions appears plausible". This is a perfect way to summarize the key findings described in the paper; and 2- "These results indicate that the synoptic processes behind the 4.2 ka BP event involved changes not only in average conditions (as reported by the speleothem) but also significant changes at the seasonal scale". The last sentence summarizes what will probably be a key research avenue for most palaeo-environmentalists during the coming years. The article could be published as it stands but I strongly encourage its publication after minor revisions.

We thank referee#1 for this general conclusion

Attached the version with tracked changes

Please also note the supplement to this comment:
https://www.clim-past-discuss.net/cp-2018-127/cp-2018-127-AC1-supplement.pdf

**Supplement:**

[revised manuscript text omitted]

---

## Author Comment (AC2) · 14 Dec 2018

We thank referee # 2 for the comments and suggestions. Following our replay point by point:

"The study provides a new data-series from Corchia Cave, focusing on the period of the 4.2 ka BP event. Although Corchia cave, together with Renella Cave, have provided multiple and significant proxies until now, this addition is of great interest since it is appropriate for the description and the exact timing of the aforementioned event, as a matter of resolution and dating. The article is well structured and well written, helping the reader to follow the discussion all along the text. The figures are helpful

and well presented. The authoring team has done a good job to provide another useful dataset, this time from the Holocene of the Central Mediterranean. Below my list of recommendations as improvements of the article"

We are happy of the positive general comment

"Title: Why 'speleothems'? I was expecting to see more than one after reading the title. Isn't it a bit misleading?"

Yes, right. Correction done.

Page 2, line 4 Gulf of Genoa, please follow the same term in whole text

Yes, right. Correction done.

"Page 2, Section 2 I find it a bit difficult to follow the site description without being able to visualize the cave setting and the general location in a large scale map. There is no need for a cave description, but I think figure 1 itself cannot help the reader much. I would suggest adding a simple map of the location and the cave setting, incl. the sampling site."

We added a new figure (Fig. 2), which includes all the requested information.

"Page 2, Section 3 It is necessary to provide a figure with the speleothem, the subsampling positions for all analyses and the projected growth axis (incl. length or distance from top)."

This has been included in the new figure 2.

"Page 3, Section 2 (line 1-5) Please, provide also the preparative method of the stable isotope analysis (e.g. acid digestion, including either a citation or a short description with acid type, temperature and duration)."

We added the requested information

"Please also provide the carbon and oxygen isotope composition of the internal standard NEW1, including their uncertainties or SD."

We added the requested information

"Page 5, Section 5.1.1 It is common to use the Mg/Ca ratio in speleothem studies, as the authors did in their previous work as well. Why here only Mg is used, without considering the oscillations of the Ca in these layers? In Fig. 4, we can see Mg/Ca data from CC27 and they look milder as a matter of excursions. To my understanding, Mg/Ca provides a more reliable proxy for interpretation. Nevertheless, Mg/Ca is not discussed, although presented in Fig. 4 (see for instance Page 6, line 31)."

Yes, we agree LA-ICP-MS cannot give reliable Mg/Ca. Specifically for Fig. 5 (in the new numbering) for a more detailed comparison with Renella speleothem RL4 (for which concentration has been meadured using standard inductively coupled plasma atomic emission spectroscopy) so we assumed to constant concentration of Ca. However, the Ca concentration is usually so high if compared to other minor/trace element that the ratio is not significantly affected. A note was inserted in the text.

"Page 6, line 25 '. . .prominent drier interval'. Here, there is a characterization of a drier event without some supportive remarks. There should be a simple statistical analysis in order to point out the significant events along the CC27 dataset. The statistical analysis can support the interpretation regarding the prominent climatic oscillations/events (for instance, which intervals are prominent indeed and which are not) and help the authors support their conclusions."

We have added a discussion on square-wave plot vs age of index of filtered anomalies, which shows that between ca. 4.5 and 4.1 there is a statistically significant interval interpreted as drier.

"References Please check some citations in the text (for example page 9, line 22 and line 33), where 'et al.' is missing. Also check please the alphabetical order of the reference list (eg. page 21, line 28).

Yes, done

"Figure 5. Proxies b, c, j and k datasets are of significantly lower resolution in comparison with the data of CC27 and the purpose of this study (4.2 ka BP event). Do they really help in this figure to make a point out of it?

We partially agree. The meaning of this figure is to try to understand which kind of climate variability can be found in archives close to the Corchia cave

Based on the recommendations-suggestions above for the improvement of the article, I would suggest the paper to be accepted for publication subject to minor revisions"

Attached the tracked changes version

Please also note the supplement to this comment:
https://www.clim-past-discuss.net/cp-2018-127/cp-2018-127-AC2-supplement.pdf

[Figure]

**Supplement:**

[revised manuscript text omitted]